# Seg-ReSearch: Segmentation with Interleaved Reasoning and External Search

**Tianming Liang** [1]    **Qirui Du** [1]    **Jian-Fang Hu** [1 2 3]    **Haichao Jiang** [1]    **Zicheng Lin** [1]    **Wei-Shi Zheng** [1 3]

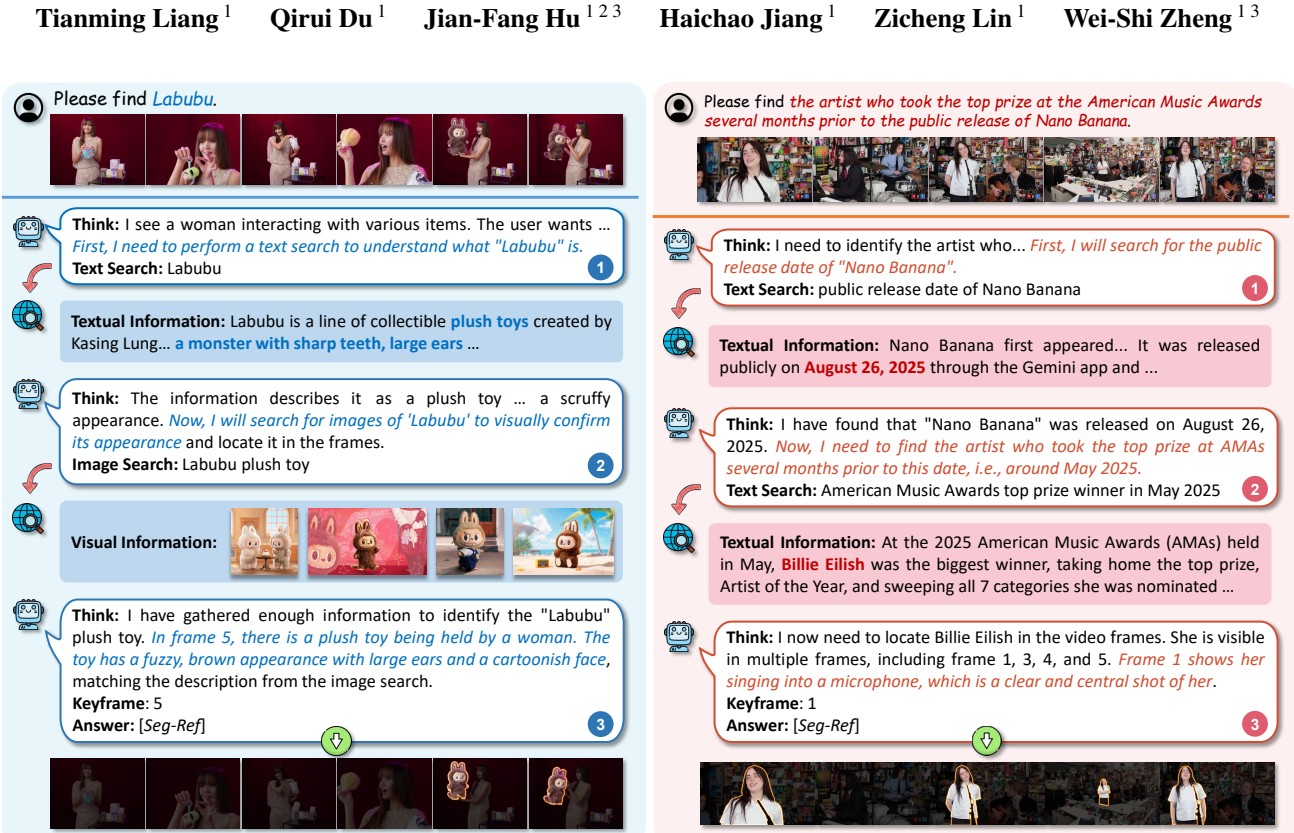

(a) Segmentation for new concepts

(b) Segmentation with up-to-date information

*Figure 1.* Through multi-turn interleaved reasoning and web search, our Seg-ReSearch is able to localize and segment any language-referred target in videos, even those involving new concepts or up-to-date information that lies beyond the internal knowledge of MLLMs.

## Abstract

Segmentation based on language has been a popular topic in computer vision. While recent advances in multimodal large language models (MLLMs) have endowed segmentation systems with reasoning capabilities, these efforts remain confined by the frozen internal knowledge of MLLMs, which limits their potential for real-world scenarios that involve up-to-date information or domain-specific concepts. In this work, we propose **Seg-ReSearch**, a novel segmentation paradigm that overcomes the knowledge bottleneck of existing approaches. By enabling interleaved reasoning and external search, Seg-ReSearch empowers segmentation systems to handle dynamic, open-world queries that extend beyond the frozen knowledge of MLLMs. To effectively train this capability, we introduce a hierarchical reward design that harmonizes initial guidance with progressive incentives, mitigating the dilemma between sparse outcome signals and rigid step-wise supervision. For evaluation, we construct OK-VOS, a challenging benchmark that explicitly requires outside

[1]School of Computer Science and Engineering, Sun Yatsen University, China [2]Guangdong Province Key Laboratory of Information Security Technology, China [3]Key Laboratory of Machine Intelligence and Advanced Computing, Ministry of Education, China. Correspondence to: Jian-Fang Hu <hujf5@mail.sysu.edu.cn>.

*Proceedings of the 43rd International Conference on Machine Learning*, Seoul, South Korea. PMLR 306, 2026. Copyright 2026 by the author(s).

knowledge for video object segmentation. Experiments on OK-VOS and two existing reasoning segmentation benchmarks demonstrate that our Seg-ReSearch improves state-of-the-art approaches by a substantial margin. Code and data is available at https://github.com/iSEE-Laboratory/Seg-ReSearch.

## 1. Introduction

Localizing and segmenting the objects of interest in images or videos has been a long-standing topic in AI community. Language, as the most natural interface, is commonly used as the reference for humans to indicate the target objects. With the recent advent of multi-modal large language models (MLLMs), this language reference is evolving from explicit descriptions to implicit reasoning instructions.

However, existing research is still limited to understanding objects within the given visual context (Ding et al., 2023; Bai et al., 2024; Liang et al., 2025; Li et al., 2026; Jiang et al., 2026). In reality, we live in a dynamic world, where user queries often involve up-to-date information or domain-specific concepts that exceed the frozen knowledge of MLLMs. For example, a user may directly request to segment "*the new Tesla*" instead of "*the white vehicle on the left*". While recent works, from LISA (Lai et al., 2024) to VideoSeg-R1 (Xu et al., 2026), have endowed segmentation models with reasoning capabilities, their potential is bounded by the static internal knowledge of MLLMs, as they lack the ability to continuously acquire necessary information from external sources.

Although recent advances in text-centric domains have adopted reinforcement learning (RL) to empower LLMs with external search abilities, adapting such methods to multi-modal tasks remains challenging, primarily due to the dilemma in reward designs. Existing approaches either rely solely on ultimate outcome rewards (Li et al., 2025c; Zheng et al., 2025b; Lin et al., 2025a; Hong et al., 2025) or impose step-wise process supervision (Zheng et al., 2025a; Liu et al., 2025d; Deng et al., 2025). However, neither approach effectively balances external search and visual reasoning. The former suffers from sparse signals, often leading the agent to bypass search steps and seek visual shortcuts. Conversely, the latter tends to over-prioritize the imitation of the search trajectory and thus neglect visual reasoning.

In this work, we propose **Seg-ReSearch**, a novel agentic **Seg**mentation framework capable of interleaved **Re**asoning and external **Search**. As illustrated in Figure 1, Seg-ReSearch performs task decomposition, reasoning, and interacting with search engines iteratively until it finds the target objects. This paradigm breaks the knowledge bottleneck of MLLMs, enabling segmentation systems to handle a broader range of user queries.

To effectively incentivize such capabilities, we introduce a hierarchical reward mechanism that strikes a balance between sparse outcome rewards and rigid step-wise imitation. Specifically, we decouple the process supervision into initial guidance and progressive incentives. For initial guidance, we utilize expert actions to supervise the first step, thereby providing a reasonable starting point for the subsequent exploration. For intermediate steps, rather than enforcing trajectory imitation, we use format-based rewards to encourage diverse yet valid search steps, coupled with a tapering bonus to prevent infinite search loops. This mechanism effectively bridges the gap between external search and visual reasoning, driving the evolution of the search behavior from initial exploration to efficient and accurate exploitation.

To assess the effectiveness of Seg-ReSearch, we establish **OK-VOS**, a novel language-instructed Video Object Segmentation benchmark that specifically requires Outside Knowledge. This benchmark is carefully annotated by human experts to ensure that each query contains up-to-date information or new concepts that lie beyond the internal knowledge of existing MLLMs. These queries may involve single or multi-hop searches, demanding complex reasoning across local visual contexts and external information. Our benchmark reveals that existing state-of-the-art (SOTA) reasoning segmentation models struggle significantly in such open-world scenarios. In contrast, our Seg-ReSearch demonstrates superior performance, even outperforming the baselines equipped with the same search tools by over 10 points. Furthermore, Seg-ReSearch also establishes new SOTA results on conventional reasoning segmentation benchmarks, including ReasonSeg (image) and ReasonVOS (video).

Our main contributions are summarized as follows:

- **Framework.** We propose Seg-Research, a novel trainable agentic segmentation framework capable of interleaved reasoning and external search.

- **Reward Design.** We introduce a hierarchical reward mechanism to effectively address the dilemma between sparse outcome signals and rigid process supervision.

- **Benchmark.** We establish OK-VOS, a human-annotated benchmark that explicitly requires external knowledge for video object segmentation.

- **SOTA Performance.** Our approach significantly outperforms SOTA reasoning segmentation models and the baselines using the same search tools.

## 2. Related Work

**Language-guided Segmentation.** Traditional segmentation methods typically rely on visual prompts (Rother et al., 2004; Xu et al., 2018) or predefined categories (He

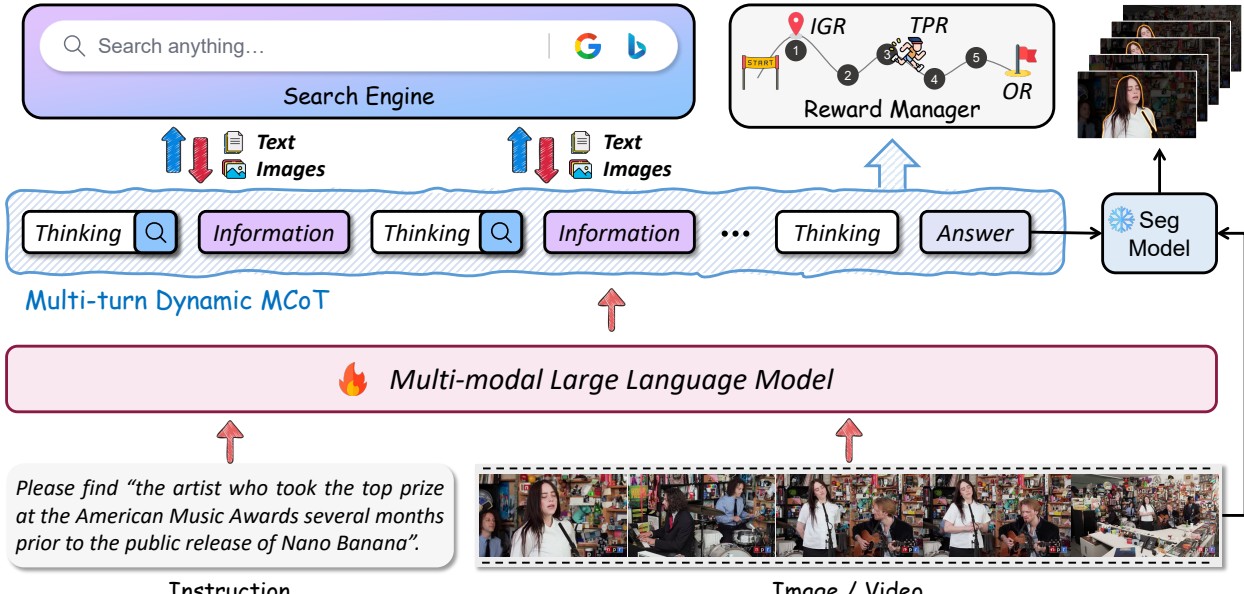

*Figure 2.* In order to identify the target objects involving new information, Seg-ReSearch conducts multi-turn interactions with the external search engine throughout the dynamic **Multi-modal Chain-of-Thought (MCoT)**. This capability is incentivized by a 3-level reward structure: **IGR** pilots the initial planning, **TPR** encourages extensive exploration, and **OR** ensures final task accuracy.

et al., 2017; Cordts et al., 2016). This task was broadened by Referring Segmentation (Kazemzadeh et al., 2014; Yu et al., 2016) and Referring Video Object Segmentation (RVOS) (Seo et al., 2020; Ding et al., 2023; Liang et al., 2026), which allow users to specify objects of interest in images or videos using natural language. To handle more implicit instructions, LISA (Lai et al., 2024) proposed Reasoning Segmentation, introducing special tokens (e.g., [SEG]) to bridge MLLMs and segmentation models. This token-based training paradigm was widely adopted in subsequent works (Ren et al., 2024; Qian et al., 2025), and further extended to video domain (Bai et al., 2024; Yan et al., 2024). To emphasize the reasoning capability, several recent efforts (Liu et al., 2025b;c) shift towards a decoupled strategy that trains the policy MLLM to generate positional prompts for an external segmentation model. This strategy facilitates reinforcement learning, incentivizing the Chain-of-Thought (CoT) capability in MLLMs (Huang et al., 2025; Xu et al., 2026). Despite these advances, current efforts remain confined to understanding the given visual context under a static reasoning paradigm. To advance this task towards more realistic scenarios, we introduce Seg-ReSearch to enable dynamic reasoning with external search, and establish OK-VOS that requires outside knowledge for segmentation.

**Reward Designs in Agentic RL.** While RL has shown promise in augmenting MLLMs with agentic abilities, designing a reward function that accurately aligns with human intent remains a significant challenge. Existing approaches typically fall into two paradigms: **(1)** sparse outcome rewards that supervise the entire trajectory (Jin et al., 2025; Su

et al., 2025; Li et al., 2025c; Zheng et al., 2025b; Hong et al., 2025; Zeng et al., 2025), and **(2)** rigid process rewards that supervise each action step (Zheng et al., 2025a; Liu et al., 2025d; Deng et al., 2025; Cheng et al., 2025; Xi et al., 2025). However, both paradigms fail to offer sufficient and flexible feedback, often leading to training instability or reward hacking. In this work, we introduce a novel hierarchical reward mechanism to address this dilemma.

**Search Agents.** LLMs are inherently limited by their fixed and outdated internal knowledge. To address this, Retrieval-Augmented Generation (RAG) systems (Izacard et al., 2023; Lewis et al., 2020) are developed, which retrieve relevant passages based on the user query and incorporate them into the LLM inputs. However, such static retrieval often suffers from noisy information. Thus, recent advances have shifted towards training LLMs to perform iterative search during reasoning (Jin et al., 2025; Zheng et al., 2025a; Li et al., 2025a;b; Geng et al., 2025). While promising, nearly all these efforts focus only on general question answering. Extending this paradigm to fine-grain visual tasks, such as segmentation, is not a minor tweak but a critical step to enable truly open-world visual understanding.

## 3. Seg-ReSearch

An overview of Seg-ReSearch is depicted in Figure 2. Given visual inputs (image or video) and a query potentially requiring external knowledge, Seg-ReSearch engages in multi-turn interactions with the search engine via a dynamic MCoT, iteratively updating its reasoning until the target

is identified. This complex capability is effectively incentivized by our hierarchical reward mechanism, which mitigates the dilemma between sparse outcome signals and rigid step-wise supervision. In the following, we elaborate on the video segmentation pipeline, as it inherently encompasses the processing required for static images.

### 3.1. Reasoning Segmentation with External Search

In this section, we detail the pipeline of Seg-ReSearch. In contrast to existing reason segmentation paradigms that depend solely on the internal knowledge of MLLMs, our method is capable of retrieving necessary external knowledge by interacting with search engines.

**Interaction with the Search Engine.** Given $N$ low-resolution video frames and a complex object query, the policy MLLM begins by analyzing the multi-modal input and planning its subsequent steps. At each step, if external knowledge is required, the model generates a search query and specifies the search tool (e.g., *search_text* or *search_image*) within the tags <search> and </search> for the search engine. The retrieved information is then wrapped inside the tags <information> and </information> and appended into the ongoing rollout sequence, serving as additional context as for subsequent generation steps. This interleaved reasoning-search process continues iteratively until the target is identified or the maximum number of search turns is reached.

**Answer Generation.** The final response of the policy MLLM comprises two stages: keyframe selection and object localization. After the multi-turn searches, the model must select a keyframe that best shows the target object and encapsulate the chosen frame index within the tags <keyframe> and </keyframe>. The corresponding high-resolution keyframe is then appended to the ongoing context. Upon that, the model localizes the target and outputs a formatted positional prompt (i.e., a bounding box and a point) within the tags <answer> and </answer>. This positional prompt, which identifies the target location on the keyframe, is passed to a frozen mask generator (e.g., SAM2 (Carion et al., 2026)) for mask prediction and propagation. For static image segmentation, we bypass the keyframe phase.

### 3.2. Hierarchical Reward Designs

Existing agentic RL approaches often struggle to balance exploration and task performance, relying either solely on sparse outcome rewards or rigid step-wise imitation. To address this dilemma, we introduce a hierarchical reward mechanism, which assesses the rollout trajectory at multiple levels, but imposes only loose constraints at each level. We find that this design can effectively achieve a trade-off between process supervision and preventing reward hacking.

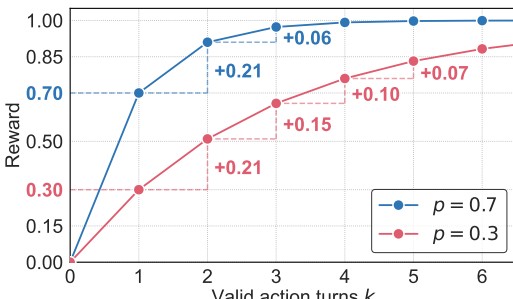

*Figure 3.* The growth curve of TPR with increasing action turns.

Formally, our reward function is defined as follows:

$$\mathcal{R} = \alpha \cdot \underbrace{(\mathcal{R}_{\text{IGR}} + \mathcal{R}_{\text{TPR}})}_{\text{Process Reward}} + \mathcal{R}_{\text{OR}}, \tag{1}$$

where $\alpha$ is a coefficient of the process reward.

**Initial Guidance Reward (IGR).** We use IGR to provide a reasonable starting point for subsequent exploration. Formally, IGR is designed as a binary reward that determines whether the first-turn generated search query $\hat{a}_0$ matches any query in the expert search trajectory $\mathcal{S}$:

$$\mathcal{R}_{\text{IGR}} = \mathbb{I}\left(\max_{s \in \mathcal{S}} \text{Sim}(\hat{a}_0, s) > 0.5\right), \tag{2}$$

where $\mathbb{I}(\cdot)$ is the indicator function and $\text{Sim}(\cdot, \cdot)$ is the semantic similarity computed by a light Sentence Transformer. Here, instead of enforcing an imitation of the expert's initial action, we allow the policy MLLM to initiate the task from any valid entry point.

**Tapering Process Reward (TPR).** Instead of forcing strict alignment with ground-truth sequences, TPR incentivizes valid exploration using only format-based rewards. Specifically, the model receives a bonus each time it outputs an action in the correct format. However, this naive strategy can lead to reward hacking, where the agent performs infinite, meaningless searches to accumulate rewards. To mitigate this, TPR incorporates a tapering bonus design, which gives the model more freedom to explore while naturally preventing it from becoming trapped in infinite loops. Formally, TPR is defined as follows:

$$\mathcal{R}_{\text{TPR}} = 1 - (1 - p)^{\min(k, M)}, \tag{3}$$

where $p \in [0, 1]$ serves as the base reward, $k$ is the number of actions following valid format (including search and answer turns), and $M$ controls the upper bound. As shown in Figure 3, a rollout sequence that directly generates answers only receives the base reward, while additional search efforts yield a convergent bonus that drives the total TPR toward 1.

**Outcome Reward (OR).** This reward is designed to capture both the quality of keyframe selection and the accuracy of

spatial localization. Formally, it comprises four terms:

$$\mathcal{R}_{\text{OR}} = \mathcal{R}_{iou} + \mathcal{R}_{l1} + \mathcal{R}_{point} + \mathcal{R}_{frame}. \quad (4)$$

Here, $\mathcal{R}_{iou}$, $\mathcal{R}_{l1}$ and $\mathcal{R}_{point}$ are *binary* rewards that measure the accuracy of the predicted bbox and point in the selected keyframe. Specifically, $\mathcal{R}_{iou}$ is assigned 1 only if the IoU between the predicted and ground-truth bounding boxes exceeds 0.5. $\mathcal{R}_{l1}$ is assigned 1 only if their $L_1$ distance is less than 10 pixels. $\mathcal{R}_{point}$ is assigned 1 only if the predicted point falls within the predicted box and its Euclidean distance to the ground-truth point is less than 100 pixels. Conversely, $\mathcal{R}_{frame}$ is a continuous reward designed to prioritize frames where the target is most prominent and least occluded. It is defined as $\mathcal{R}_{frame} = \frac{C_j}{\max_i C_i}$, where $C_i$ denotes the area of the largest connected component in the ground-truth mask of the $i$-th frame, and $j$ is the index of the selected keyframe.

### 3.3. Reinforcement Learning with a Search Engine

**Rollout Formulation.** The rollout sequence of Seg-ReSearch can be formulated as a multi-turn long-horizon trajectory interacting with the search engine. A trajectory of $T$-turn interactions is described as $\mathcal{T} = \{(a_t, e_t)\}_{t=1}^{T}$, where $a_t$ represents the text sequence (comprising reasoning and search tokens) generated by the policy MLLM at turn $t$, and $e_t$ denotes environmental feedback (i.e., the information retrieved from the search engine). Let $x$ represent the multi-modal inputs (system prompt, user instruction, and input video) and $y$ denote the entire answer sequence. The joint generation process is formulated as:

$$P_\theta(\mathcal{T}, y \mid x) = \left[ \prod_{t=1}^{T} \pi_\theta(a_t \mid \mathcal{T}_{<t}, x) \right] \cdot \pi_\theta(y \mid \mathcal{T}, x)$$

$$= \left[ \prod_{t=1}^{T} \pi_\theta(a_t \mid \mathcal{T}_{<t}, x) \cdot \mathcal{E}(e_t \mid a_t) \right] \cdot \pi_\theta(y \mid \mathcal{T}, x)$$

$$\propto \left[ \prod_{t=1}^{T} \pi_\theta(a_t \mid \mathcal{T}_{<t}, x) \right] \cdot \pi_\theta(y \mid \mathcal{T}, x) \quad (5)$$

where $\pi_\theta(\cdot \mid \mathcal{T}, x)$ represents the policy MLLM, and $\mathcal{E}(e_t \mid a_t)$ denotes the environmental transition probability, which is typically treated as constant.

**Optimization.** Seg-ReSearch is optimized following the paradigm of Group Relative Policy Optimization (GRPO). For brevity, let $o = [\mathcal{T}, y]$ represent the complete output sequence. For each input $x$, GRPO samples a group of outputs $\{o_1, o_2, \ldots, o_G\}$ from the old policy $\pi_{\theta_{old}}(o_i|x)$, evaluates their corresponding rewards $\{\mathcal{R}_1, \mathcal{R}_2, \ldots, \mathcal{R}_G\}$, and computes the advantage as follows:

$$A_i = \frac{\mathcal{R}_i - \text{mean}(\{\mathcal{R}_1, \mathcal{R}_2, \ldots, \mathcal{R}_G\})}{\text{std}(\{\mathcal{R}_1, \mathcal{R}_2, \ldots, \mathcal{R}_G\})}. \quad (6)$$

Ultimately, the policy model $\pi_\theta$ is optimized by maximizing the following objective function:

$$\mathcal{J}(\theta) = \mathbb{E}_{x \sim \mathcal{D}, \{o_i\}_{i=1}^G \sim \pi_{\theta_{old}}(\cdot|x)} \left[ \frac{1}{G} \sum_{i=1}^{G} \min \Big( \right.$$

$$\left. \frac{\pi_\theta(o_i|x)}{\pi_{\theta_{old}}(o_i|x)} A_i, \text{clip}\big(\frac{\pi_\theta(o_i|x)}{\pi_{\theta_{old}}(o_i|x)}, 1-\epsilon, 1+\epsilon\big) A_i \Big) \right], \quad (7)$$

where $\epsilon$ is the clipping threshold.

## 4. OK-VOS: A VOS Benchmark Requiring Outside Knowledge

Existing language-guided segmentation benchmarks (Seo et al., 2020; Kazemzadeh et al., 2014; Liang et al., 2026; Ding et al., 2025) always assume that the user inputs already provide all necessary evidence for identifying the target objects. While reasoning segmentation benchmarks (Lai et al., 2024; Bai et al., 2024; Yan et al., 2024) emphasize world knowledge, they tend to involve only basic common sense (e.g., "which food is rich in Vitamin C"). These simplified settings fail to reflect the real-world scenarios that often involve up-to-date information or long-tail knowledge. To bridge this gap, we establish OK-VOS, a new video object segmentation benchmark that explicitly requires outside knowledge for object identification. This benchmark is fully annotated by five human experts. It contains 1,000 test samples, covering 150 videos and 500 objects. We conduct a multi-round review and re-annotation process to strictly ensure that each query requires up-to-date information or long-tail facts that explicitly exceeds the internal knowledge of current LLMs. Any queries that could be correctly answered by LLMs without external web search were discarded or refined. Furthermore, we take rigorous measures to mitigate potential biases that allow the object targets to be easily identified through visual shortcuts. For example, queries like "*who is the 2025 Oscar winner for Best Actress*" are excluded if the video clip contains only one woman.

To comprehensively evaluate the model's capabilities in iterative search and reasoning, we categorize the samples into the following three hierarchical types. (i) **One-hop**: the target can be identified via a single, direct retrieval. (ii) **Multi-hop**: identifying the target requires retrieving and connecting multiple pieces of information. (iii) **Relational**: the target is defined by its visual relationship to another knowledge-based anchor target, e.g., "*who received the ball from the winner of 2025 European Golden Ball*". Finally, the benchmark contains 229 One-hop, 495 Multi-hop, and 276 Relational samples. By bridging the gap between fine-grained visual understanding and external information retrieval, OK-VOS establishes a crucial and challenging testbed for open-world AI systems.

*Table 1.* Comparison on OK-VOS benchmark.

| Method | One-hop | | | Multi-hop | | | Relational | | | Overall | | |
|---|---|---|---|---|---|---|---|---|---|---|---|---|
| | $\mathcal{J}\&\mathcal{F}$ | $\mathcal{J}$ | $\mathcal{F}$ | $\mathcal{J}\&\mathcal{F}$ | $\mathcal{J}$ | $\mathcal{F}$ | $\mathcal{J}\&\mathcal{F}$ | $\mathcal{J}$ | $\mathcal{F}$ | $\mathcal{J}\&\mathcal{F}$ | $\mathcal{J}$ | $\mathcal{F}$ |
| ***Specialist RVOS models*** | | | | | | | | | | | | |
| SAMWISE CVPR'25 | 27.0 | 25.8 | 28.2 | 26.1 | 25.4 | 26.8 | 24.7 | 23.3 | 26.0 | 25.9 | 24.9 | 26.9 |
| ReferDINO ICCV'25 | 25.1 | 24.2 | 26.0 | 27.7 | 26.8 | 28.6 | 18.7 | 17.4 | 20.1 | 24.6 | 23.6 | 25.6 |
| ReferEverything-1.4B ICCV'25 | 25.0 | 24.0 | 26.1 | 27.7 | 27.4 | 28.1 | 24.3 | 23.1 | 25.5 | 26.2 | 25.4 | 26.9 |
| ***MLLM-based methods*** | | | | | | | | | | | | |
| VideoLISA-3.8B NeurIPS'24 | 22.9 | 22.9 | 23.0 | 21.7 | 21.8 | 21.5 | 12.0 | 11.0 | 13.0 | 19.3 | 19.1 | 19.5 |
| GLUS-7B CVPR'25 | 32.4 | 32.0 | 32.8 | 29.9 | 29.3 | 30.5 | 26.8 | 26.0 | 27.6 | 29.6 | 29.0 | 30.2 |
| RGA3-7B ICCV'25 | 22.3 | 22.9 | 21.8 | 21.0 | 21.1 | 20.9 | 15.1 | 13.9 | 16.2 | 19.7 | 19.5 | 19.8 |
| UniPixel-7B NeurIPS'25 | 37.0 | 36.6 | 37.5 | 33.4 | 32.7 | 34.1 | 33.4 | 32.2 | 34.5 | 34.2 | 33.5 | 35.0 |
| Qwen3-VL-4B* | 32.6 | 32.3 | 32.9 | 33.3 | 33.0 | 33.6 | 35.6 | 34.5 | 36.6 | 33.8 | 33.3 | 34.3 |
| Qwen3-VL-4B*+*Search* | 39.8 | 39.7 | 39.8 | 34.3 | 34.2 | 34.4 | 36.6 | 35.9 | 37.2 | 36.2 | 36.0 | 36.4 |
| **Seg-ReSearch-4B (ours)** | **54.0** | **53.6** | **54.3** | **43.3** | **42.8** | **43.9** | **44.2** | **43.0** | **45.4** | **46.0** | **45.3** | **46.7** |
| Qwen3-VL-8B* | 37.1 | 36.5 | 37.7 | 35.2 | 34.7 | 35.7 | 35.7 | 34.6 | 36.8 | 35.8 | 35.1 | 36.4 |
| Qwen3-VL-8B*+*Search* | 40.2 | 39.8 | 40.6 | 36.4 | 35.9 | 36.9 | 37.6 | 36.6 | 38.6 | 37.6 | 37.0 | 38.2 |
| **Seg-ReSearch-8B (ours)** | **60.1** | **59.7** | **60.5** | **48.3** | **47.9** | **48.6** | **44.8** | **43.6** | **46.0** | **50.0** | **49.4** | **50.7** |

## 5. Experiments

### 5.1. Experimental Setup

**Implementation Details.** We adopt Qwen3-VL-Instruct-4B and 8B (Bai et al., 2025) as our base MLLMs and manually annotate 100 samples for RL training. During training, we use E5 (Wang et al., 2022) as the retriever. Because our samples involve extensive up-to-date information that extends beyond the scope of public knowledge bases like Wikipedia Dumps (Karpukhin et al., 2020), we customize a knowledge base based on the 100 training samples. During validation, we directly invoke the Google Search API. We set the number of retrieved passages to 3, retrieved images to 1, and the maximum search turns to 5. We empirically set $\alpha = 0.5$, $p = 0.7$, $M$=10 and $\epsilon = 0.2$. All input frames and retrieved images are resized to $448 \times 448$, while the selected keyframes are resized to $864 \times 864$. For image segmentation tasks, we resize all images to $864 \times 864$. We apply SAM2 (Ravi et al., 2025) to produce segmentation masks for ReasonSeg and ReasonVOS. For OK-VOS, which involves frequent shot changes, we employ SeC (Zhang et al., 2025). More details are present in the Supplementary.

**Evaluation Metrics.** Our metrics follow the previous segmentation works. For video object segmentation, we use region similarity ($\mathcal{J}$), contour accuracy($\mathcal{F}$), and their average value ($\mathcal{J}\&\mathcal{F}$). For static image segmentation, we use generalized IoU (gIoU) and cumulative IoU (cIoU).

**Baselines.** We implement two baselines for fair comparisons: Qwen3-VL* employs the same MLLM and mask generator as ours, and Qwen3-VL*+*Search* is further equipped with the same search API and prompts. We also benchmark against existing SOTAs, including specialist RVOS models (Cuttano et al., 2025; Liang et al., 2025; Bagchi et al.,

*Table 2.* Comparison on ReasonSeg benchmark.

| Method | Val | | Test | |
|---|---|---|---|---|
| | gIoU | cIoU | gIoU | cIoU |
| LISA-7B CVPR'24 | 53.6 | 52.3 | 48.8 | 47.1 |
| RSVP-Qwen-7B ACL'25 | 58.6 | 48.5 | 56.6 | 51.6 |
| Seg-Zero-7B arXiv'25 | 62.6 | 62.0 | 57.5 | 52.0 |
| SAM-R1-7B NeurIPS'25 | 64.0 | 55.8 | 60.2 | 54.3 |
| SAM3-Agent-7B ICLR'26 | 65.4 | 50.5 | 62.6 | 56.2 |
| Qwen3-VL-8B* | 70.3 | 70.0 | 66.0 | 53.7 |
| **Seg-ReSearch-8B (ours)** | **73.3** | **72.2** | **67.4** | **59.0** |

*Table 3.* Comparison on ReasonVOS benchmark.

| Method | $\mathcal{J}\&\mathcal{F}$ | $\mathcal{J}$ | $\mathcal{F}$ |
|---|---|---|---|
| VideoLISA-3.8B NeurIPS'24 | 47.5 | 45.1 | 49.9 |
| GLUS-7B CVPR'25 | 49.9 | 47.5 | 52.4 |
| RGA3-7B ICCV'25 | 53.6 | 51.3 | 56.0 |
| OneThinker-8B CVPR'26 | 54.9 | 51.1 | 58.7 |
| Qwen3-VL-8B* | 56.9 | 53.8 | 60.0 |
| **Seg-ReSearch-8B (ours)** | **63.2** | **60.2** | **66.2** |

2025) and MLLM-based segmentation approaches (Bai et al., 2024; Lin et al., 2025b; Wang et al., 2025; Liu et al., 2025a). More details are provided in the Supplementary.

### 5.2. SOTA Comparisons

We present the comparison results on the OK-VOS benchmark in Table 1. *First*, this task poses severe challenges to existing SOTA methods, whether relying on specialist RVOS models or MLLMs. For instance, VideoLISA-3.8B achieves only 19.3 in overall $\mathcal{J}\&\mathcal{F}$, and even the recent UniPixel-7B achieves only 34.2. *Second*, simply equipping MLLMs with the search engine is insufficient. For example, Qwen3-VL-8B*+*Search* improves over the base model by only 1.8%.

Table 4. $\mathcal{J}\&\mathcal{F}$ results of various reward designs. Here, *Baseline* indicates the Qwen3-VL-4B*+*Search*.

| Rewards | One-hop | Multi-hop | Relational | Overall |
|---|---|---|---|---|
| Sparse | 51.0 | 39.3 | 36.7 | 41.3 |
| Rigid | 51.0 | 41.8 | 40.7 | 43.6 |
| **Ours** | **54.0** | **43.3** | **44.2** | **46.0** |

Table 5. Effects of IGR and TPR.

| IGR | TPR | $\mathcal{J}\&\mathcal{F}$ | $\mathcal{J}$ | $\mathcal{F}$ |
|---|---|---|---|---|
| | | 41.3 | 40.6 | 41.9 |
| ✓ | | 42.0 | 41.3 | 42.7 |
| | ✓ | 44.8 | 44.2 | 45.4 |
| ✓ | ✓ | **46.0** | **45.3** | **46.7** |

Table 6. Overall performance across different $p$ values.

| $p$ | $\mathcal{J}\&\mathcal{F}$ | $\mathcal{J}$ | $\mathcal{F}$ |
|---|---|---|---|
| 0.2 | 44.4 | 43.7 | 45.1 |
| 0.5 | 45.3 | 44.8 | 45.9 |
| 0.7 | 46.0 | 45.3 | 46.7 |
| 0.9 | 41.0 | 40.3 | 41.6 |

*In contrast*, Seg-ReSearch demonstrates substantial superiority over these competitors. Our 4B model outperforms the search-augmented baseline by nearly 10%, and our 8B model establishes a significantly superior SOTA with 50.0 in overall $\mathcal{J}\&\mathcal{F}$, validating the effectiveness of the interleved reasoning and search capabilities of Seg-ReSearch.

To demonstrate that Seg-ReSearch is also competitive in conventional reasoning segmentation tasks, we evaluate it on the ReasonSeg image benchmark and the ReasonVOS video benchmark. As shown in Table 2 and Table 3, Seg-ReSearch-8B establishes new SOTA results on both benchmarks. Notably, compared to OneThinker-8B that uses the same MLLM, our Seg-ReSearch achieves a substantial improvement of +8.3 $\mathcal{J}\&\mathcal{F}$. These results confirm the robustness of Seg-ReSearch in general reasoning segmentation.

### 5.3. Qualitative Comparison

We present a multi-hop example in Figure 4, where the user asks to identify "the SNL host on the specific date when Michaela Benthaus traveled to space." Qwen3-VL* fails to respond, as this query involves specific information that lies beyond its internal knowledge. Although Qwen3-VL*+*Search* is augmented with search tools, it tends to naively forward the raw user query to the search engine, leading to irrelevant results that hinder subsequent reasoning. In contrast, our Seg-ReSearch successfully identifies the target object by decomposing the task into iterative reasoning and external search: it first searches for "the date of the space travel" and subsequently searches for "the SNL host on that specific date". These results demonstrate the superior reasoning-search capability of Seg-ReSearch in handling complex and open-world scenarios.

### 5.4. Analysis of Reward Designs

In this section, we delve into the impacts of various reward designs on the learning dynamics and final performance. In this section, we use Qwen3-VL-4B as the base model.

**Hierarchical Reward Design.** In Table 4, we compare our hierarchical reward design against two common paradigms: *sparse* outcome rewards and *rigid* step-wise imitation. First, the outcome-based reward yields the lowest performance, indicating that sparse outcome signals fail to guide complex multi-turn reasoning. While introducing step-wise imitation improves performance, it constrains the model's exploration and leads to a sub-optimal policy. In contrast, our hierar-

chical reward design achieves the best performance. This demonstrates that our hierarchical reward design effectively balances guidance and exploration, facilitating the learning of robust, interleaved reasoning-and-search policies.

**Effects of IGR and TPR.** In Table 5, we verify the individual contribution of IGR and TPR. Interestingly, TPR serves as a "free lunch", as it boosts performance by +3.5% without requiring expert annotation. Furthermore, the combination of IGR and TPR leads to a remarkable improvement of +4.7%, demonstrating the synergy between initial guidance and progressive incentives.

**Effect of Base Reward.** We provide the results across various $p$ (i.e., the base reward of TPR) in Table 6. A larger $p$ implies a smaller bonus for extra search. With increasing from 0.2 to 0.7, the overall $\mathcal{J}\&\mathcal{F}$ increases from 44.4 to 46.0. This reflects a trade-off between extensive exploration and precise search. However, when is further increased to 0.9, the bonus becomes too marginal to encourage necessary search, causing a sharp performance drop to 41.0.

**Data Efficiency.** We analyze the impact of training data size in Table 7. Seg-ReSearch demonstrates consistent performance gains with increased training samples. In contrast, removing our process rewards (w/o PR) leads to training instability and marginal gains. This confirms the stability and scalability of our hierarchical reward mechanism.

**Learning Dynamics of Search Ability.** We compare our process reward with three variants: (1) LPR, a linear reward that gives a fixed bonus for each valid search; (2) BPR, a binary reward awarded only if all searches are valid; and (3) w/o PR, without using any process supervision. As shown in Figure 5, LPR exhibits severe reward hacking, where the policy model rapidly learns to abuse the search and degrades $\mathcal{J}\&\mathcal{F}$ to 41.1. Conversely, search usage for BPR and w/o PR are nearly unchanged during training, indicating that such sparse signals are insufficient to motivate multi-step searching. Our design effectively addresses this dilemma: it incentivizes valid exploration while preventing infinite

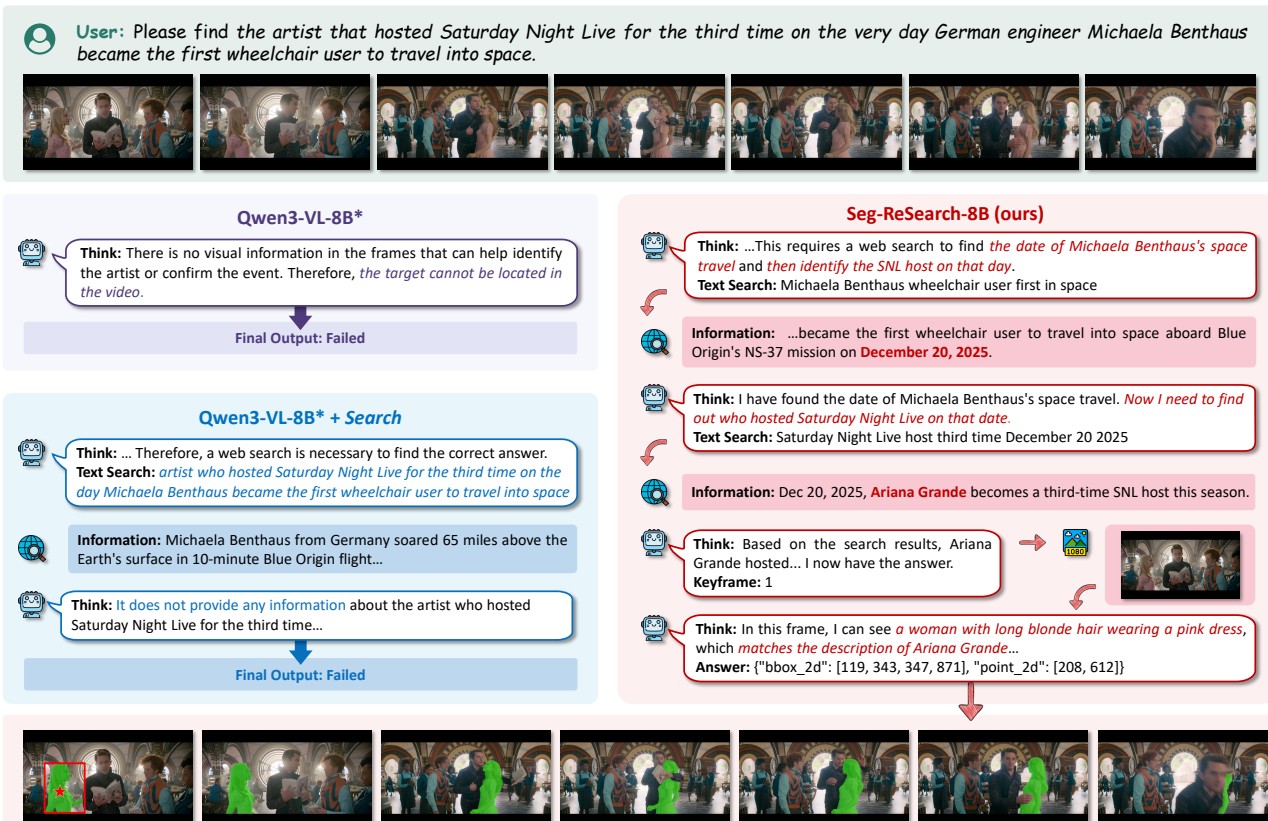

*Figure 4.* Qualitative comparison between our Seg-ReSearch and the baselines: Qwen3-VL-8B* and Qwen3-VL-8B*+*Search*.

*Table 7.* Overall $\mathcal{J}\&\mathcal{F}$ over different number of training samples.

| Method | 25 | 50 | 75 | 100 |
|---|---|---|---|---|
| **Seg-ReSearch** | **43.0** | **44.3** | **45.5** | **46.0** |
| w/o PR | 41.1 | 41.7 | 38.5 | 41.3 |

*Table 8.* Impact of maximum search turns.

| Max. Turns | $\mathcal{J}\&\mathcal{F}$ | $\mathcal{J}$ | $\mathcal{F}$ |
|---|---|---|---|
| 1 | 42.6 | 42.2 | 43.1 |
| 5 | 50.0 | 49.4 | 50.7 |
| 10 | 50.4 | 49.7 | 51.1 |

*Table 9.* Impact of retrieval numbers for text and images.

| Text | Image | $\mathcal{J}\&\mathcal{F}$ | $\mathcal{J}$ | $\mathcal{F}$ |
|---|---|---|---|---|
| 1 | 1 | 47.9 | 47.3 | 48.6 |
| 3 | 1 | 50.0 | 49.4 | 50.7 |
| 3 | 3 | 50.8 | 50.2 | 51.4 |

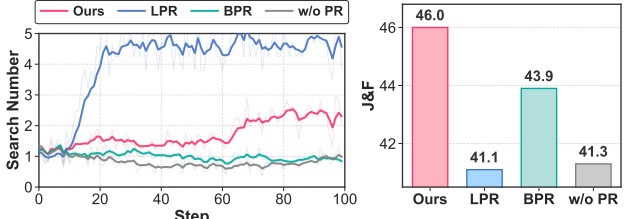

*Figure 5.* Left: number of search calls over training steps. Right: overall performance of various methods.

loops, steadily increasing search turns to ∼2.5 and achieving the best performance of 46.0 $\mathcal{J}\&\mathcal{F}$.

### 5.5. Ablation on Search Settings

We empirically study the impacts of various search settings. All experiments in this section are based on Seg-ReSearch-8B. Default settings are highlighted in the tables.

**Maximum Search Turns.** We study the impacts of varing maximum action budget in Table 8. The performance jumps dramatically by +7.4% when increasing the limit from 1 to 5 turns, demonstrating the necessity of multi-turn reasoning and search. However, increasing the budget further to 10 turns results in a marginal gain of 0.4%, highlighting the efficient and accurate search capability of our Seg-ReSearch.

**Retrieval Number.** In Table 9, we analyze the impact of the retrieval number (i.e., Top-$k$) per search. Increasing the number of textual entries from 1 to 3 significantly boosts performance by +2.1%, implying that textual search serves

*Table 10.* Overall $\mathcal{J}\&\mathcal{F}$ over different search engines

| Search Engine | $\mathcal{J}\&\mathcal{F}$ | $\mathcal{J}$ | $\mathcal{F}$ |
|---|---|---|---|
| DuckDuckGo | 47.1 | 46.4 | 47.8 |
| Google | 50.0 | 49.4 | 50.7 |
| Google+Browsing | 54.4 | 53.8 | 55.0 |

*Table 11.* Average time cost (sec. / sample) on various benchmarks.

| Method | $\mathcal{J}\&\mathcal{F}$ | $\mathcal{J}$ | $\mathcal{F}$ |
|---|---|---|---|
| Qwen3-VL-8B* | 3.2 | 1.7 | 6.1 |
| **Seg-ReSearch** | 4.1 | 1.8 | 8.0 |

as the primary source of external knowledge. Furthermore, increasing image retrieval yields an additional 0.8% gain, demonstrating the complementary benefit of multi-modal search. To strike a balance between accuracy and efficiency, we adopt 3 textual and 1 visual entry as our default setting.

**Search Engines.** We compare the performance of using different search backends. As shown in Table 10, Google search surpasses DuckDuckGo by 2.9 $\mathcal{J}\&\mathcal{F}$. Therefore, we use Google Search by default. Interestingly, we further explore the potential of web browsing, where we access the full content of each web page and employ an additional LLM for summarization. We find that this strategy yields a substantial gain of +4.4 $\mathcal{J}\&\mathcal{F}$. This indicates that the potential of Seg-ReSearch can be further broadened by constructing a more comprehensive retrieval system.

**Time Cost.** As shown in Table 11, for queries that do not require external knowledge (e.g., ReasonSeg and Reason-VOS), Seg-ReSearch only brings a negligible extra time burden (<1s), as very few external searches are called. For complex queries (e.g., in OK-VOS) that require external knowledge, the multi-round search incur acceptable extra latency (~2s).

## 6. Conclusion

In this work, we propose Seg-ReSearch to break the knowledge bottleneck of existing segmentation systems. By interleaving reasoning with external search, Seg-ReSearch is able to handle dynamic, open-world queries exceeding the frozen knowledge of MLLMs. To this end, we design a hierarchical reward mechanism that effectively balances outcome feedbacks with step-wise supervision. Furthermore, we established OK-VOS, a challenging VOS benchmark that explicitly requires external knowledge. Extensive experiments demonstrate that Seg-ReSearch achieves substantial improvements over existing approaches. We hope this work serves as a solid step towards building more autonomous and knowledgeable visual agents capable of continuously interacting with the dynamic world.

## Acknowledgement

This work was supported partially by the NSFC (U22A2095, 62476296), Guangdong Natural Science Funds Project (2023B1515040025, 2022B1111010002), Guangdong NSF for DistinguishedYoung Scholar (2022B1515020009).

## Impact Statement

This paper presents a significant step towards universal visual recognition and segmentation systems by enabling models to access the external world. The proposed Seg-ReSearch holds great promise for next-generation visual intelligent agents, enabling users to interact with visual content more intuitively and acquire deeper insights beyond pixel-level perception. This technique can also be applied to robotic and educational scenarios to improve the open-world and dynamic abilities of assistant systems. On the negative side, it may introduce potential risks such as amplifying internet bias and raising privacy concerns. Nevertheless, considering the broader perspective, we believe this work is meaningful for advancing the machine learning community towards real-world artificial intelligence. Therefore, we advocate for responsible development, and believe its contributions significantly outweigh the potential risks.

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

# A. More Implementation Details

Following Search-R1 (Jin et al., 2025), we mask retrieved tokens during loss computation, ensuring that policy gradients are calculated solely based on MLLM-generated tokens. Training is conducted on a single node with 8 NVIDIA A6000 GPUs. We use a global batch size of 16 and a mini-batch size of 8. The maximum response length is set to 6,144 tokens, comprising 4,096 tokens for generation and 2,048 tokens for environmental feedback. For IGR, we use a lightweight sentence transformer all-MiniLM-L6-v2 (Wang et al., 2020) to compute the semantic similarity. We train Seg-ReSearch-4B for 100 epochs without KL penalty, and Seg-ReSearch-8B for 120 epochs with a KL divergence coefficient of 0.01. The training set comprises 100 samples, consisting of 80 samples requiring search and 20 samples without search. For each video, we uniformly sample 6 frames as input. While we believe more sophisticated frame selection strategies (Yan et al., 2024; Yu et al., 2025) might further enhance performance, exploring them is beyond the scope of this work. The system prompt used for training is provided in the following.

---

**System Prompt for Seg-ReSearch**

Your role is a video target identification assistant capable of web search. You will be given an object query and a sequence of video frames. Each frame is preceded by its index. Your task is to locate the target with a bounding box and a point in your selected frame.

# Tools
You have access to the following tools:
<tools>
{"name": "text_search", "description": " Search for textual information from the internet.", "parameters": {"query": {"type": "string", "description": "Search keywords or phrases"}}
{"name": "image_search", "description": " Search for images from the internet.", "parameters": {"query": {"type": "string", "description": "Search keywords or phrases"}}
</tools>
Each tool will return the top searched results between <information> and </information>.

# Output Format:
Depending on the situation, output one of the following:

## 0. If you need web search (zero to multiple times):
<think> (1) Plan your reasoning steps; (2) Justify the need for a web search. </think>
<search> {"name": <tool-name>, "query": <string>} </search>

## 1. Think deeply based on the query and the video:
<think> (1) Carefully compare all possible objects in the video and find the object that most matches the query; (2) Analyze the provided frames and select the best one where the target is most clearly visible. </think>
<keyframe> [integer frame_index] </keyframe>

## 2. Once you receive the high-res keyframe:
<think> (1) Describe the unique visual features of the target to prove you captured it; (2) Determine the precise 2D location of the target with bbox and the point inside the object. </think>
<answer> {"bbox_2d": [x1,y1,x2,y2], "point_2d": [x,y]} </answer>

# IMPORTANT NOTE
At each step, 1) always look at the video frames first before you decide to search; 2) never search for information that can be obtained from the given video; 3) do not use search if you only wanna analyze the frames.

---

**Baselines:** In this work, we mainly compare against the following SOTA models:
(1) Specialist RVOS models: SAMWISE (Cuttano et al., 2025), ReferDINO (Liang et al., 2025) and ReferEverything (Bagchi et al., 2025).
(2) MLLM-based methods for video segmentation: VideoLISA (Bai et al., 2024), GLUS (Lin et al., 2025b), RGA3 (Wang et al., 2025), UniPixel (Liu et al., 2025a), and OneThinker (Feng et al., 2026).
(3) MLLM-based methods for image segmentation: LISA (Lai et al., 2024), RSVP (Lu et al., 2025), Seg-Zero (Liu et al., 2025b), SAM-R1 (Huang et al., 2025), SAM3-Agent (**?**).

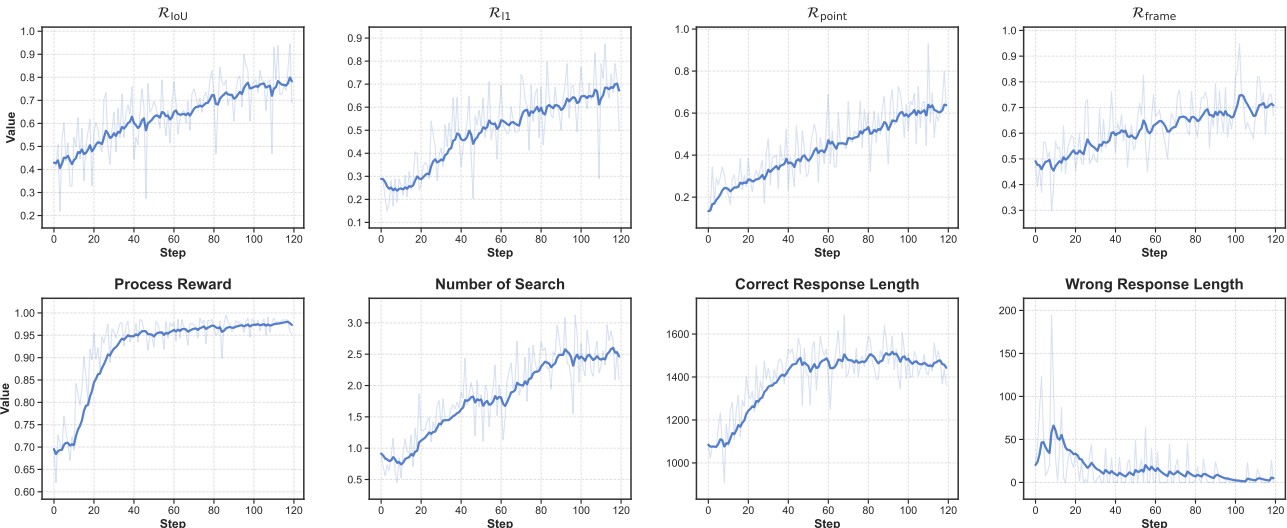

*Figure 6.* Training dynamics of Seg-ReSerach-8B.

## B. Training Dynamics of Seg-ReSearch

We visualize the evolution of Seg-ReSearch-8B during training in Figure 6. As shown in the top row, the outcome rewards (e.g., $\mathcal{R}_{iou}$, $\mathcal{R}_{l1}$, $\mathcal{R}_{point}$, and $\mathcal{R}_{frame}$) exhibit a steady upward trend. The process reward rises rapidly in the early stages, coinciding with a sharp decline in wrong response length to near zero. This demonstrates that our hierarchical reward design effectively guides the model to strictly adhere to the required interaction format before optimizing for task accuracy. Notably, the average number of searches gradually increases from 0.8 to stabilize around 2.5, reflecting the evolution of Seg-ReSearch's search ability from initial exploration to efficient, accurate exploitation.

## C. Comparison with SFT Baselines

A critical question is whether our performance gains come from extra data (the 100 samples) or our training paradigm. To answer this, we fine-tune existing SOTA models on the same 100 samples following their public supervised fine-turning (SFT) paradigm. As shown in Table 12, relying solely on SFT is insufficient to address our task. First of all, SFT is data-hungry, typically requiring extensive annotations to prevent overfitting. For example, it degrade the performance of GLUS-7B by 13.1% in overall $\mathcal{J}\&\mathcal{F}$. In addition, SFT leads to memorization, failing to teach the model how to actively use search engines to reason about unknown targets. For instance, while SFT on UniPixel-7B shows a slight improvement (+2.8%), it still lags far behind our method. In contrast, our approach is data-efficient, achieving 9.8% and 12.4% performance improvements over the 4B and 8B baselines with only 100 samples.

*Table 12.* Comparison with SFT baselines on the OK-VOS benchmark.

| Method | One-hop | | | Multi-hop | | | Relational | | | Overall | | |
|---|---|---|---|---|---|---|---|---|---|---|---|---|
| | $\mathcal{J}\&\mathcal{F}$ | $\mathcal{J}$ | $\mathcal{F}$ | $\mathcal{J}\&\mathcal{F}$ | $\mathcal{J}$ | $\mathcal{F}$ | $\mathcal{J}\&\mathcal{F}$ | $\mathcal{J}$ | $\mathcal{F}$ | $\mathcal{J}\&\mathcal{F}$ | $\mathcal{J}$ | $\mathcal{F}$ |
| GLUS-7B CVPR'25 | 32.4 | 32.0 | 32.8 | 29.9 | 29.3 | 30.5 | 26.8 | 26.0 | 27.6 | 29.6 | 29.0 | 30.2 |
| ↪ *w/ SFT* | 18.8 | 18.0 | 19.5 | 18.0 | 17.2 | 18.9 | 12.0 | 10.7 | 13.3 | 16.5 | 15.6 | 17.5 |
| Δ | **-13.6** | **-14.0** | **-13.3** | **-11.9** | **-12.1** | **-11.6** | **-14.8** | **-15.3** | **-14.3** | **-13.1** | **-13.4** | **-12.7** |
| UniPixel-7B NeurIPS'25 | 37.0 | 36.6 | 37.5 | 33.4 | 32.7 | 34.1 | 33.4 | 32.2 | 34.5 | 34.2 | 33.5 | 35.0 |
| ↪ *w/ SFT* | 39.3 | 38.8 | 39.8 | 34.9 | 34.0 | 35.8 | 39.0 | 38.2 | 39.9 | 37.0 | 36.3 | 37.8 |
| Δ | **+2.3** | **+2.2** | **+2.3** | **+1.5** | **+1.3** | **+1.7** | **+5.6** | **+6.0** | **+5.4** | **+2.8** | **+2.8** | **+2.8** |
| Qwen3-VL-4B*+*Search* | 39.8 | 39.7 | 39.8 | 34.3 | 34.2 | 34.4 | 36.6 | 35.9 | 37.2 | 36.2 | 36.0 | 36.4 |
| **Seg-ReSearch-4B (ours)** | **54.0** | **53.6** | **54.3** | **43.3** | **42.8** | **43.9** | **44.2** | **43.0** | **45.4** | **46.0** | **45.3** | **46.7** |
| Δ | **+14.2** | **+13.9** | **+14.5** | **+9.0** | **+8.6** | **+9.5** | **+7.6** | **+7.1** | **+8.2** | **+9.8** | **+9.3** | **+10.3** |
| Qwen3-VL-8B*+*Search* | 40.2 | 39.8 | 40.6 | 36.4 | 35.9 | 36.9 | 37.6 | 36.6 | 38.6 | 37.6 | 37.0 | 38.2 |
| **Seg-ReSearch-8B (ours)** | **60.1** | **59.7** | **60.5** | **48.3** | **47.9** | **48.6** | **44.8** | **43.6** | **46.0** | **50.0** | **49.4** | **50.7** |
| Δ | **+19.9** | **+19.9** | **+19.9** | **+11.9** | **+12.0** | **+11.7** | **+7.2** | **+7.0** | **+7.4** | **+12.4** | **+12.4** | **+12.5** |

## D. More Qualitative Analysis

We visualize the detailed reasoning process of Qwen3-VL-8B*+*Search* and our Seg-ReSearch in Table 13. In this sample, Qwen3-VL-8B*+*Search* fails to recognize the need for external information. Instead, it suffers from hallucination, incorrectly assuming the prominent female singer is the target. Conversely, Seg-ReSearch retrieves the specific band info via text search and verifies the singer's identity via image search, successfully identifying the target in the provided video.

*Table 13*. Visualization of the CoTs between Qwen3-VL-8B*+*Search* and our Seg-ReSearch.

**User Input:** A band is scheduled to release its third studio album, "Boycott Heaven," in 2026. Please find the lead singer of this band in the provided video.

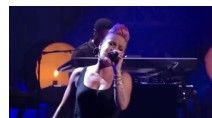 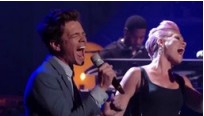 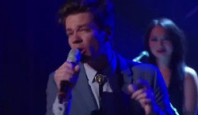 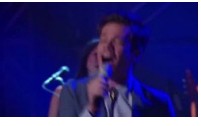 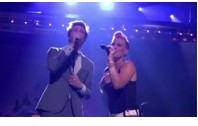 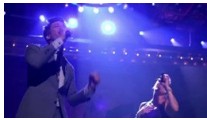

**Qwen3-VL-8B*+*Search*:**

<think> The video frames show a performance with two main singers, a man and a woman. The woman has distinctive pink hair and is singing into a microphone in multiple frames, suggesting she is the lead singer. The man is also singing but appears to be a supporting vocalist. Therefore, the woman with pink hair is most likely the lead singer. </think>
<keyframe> 1 </keyframe>

Here is the high-res version of your selected frame:

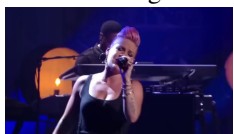

<think> The woman in the frame has distinctive pink hair, is wearing a black top, and is holding a microphone while singing. These are clear visual features that identify her as the lead singer. </think>
<answer> {"bbox_2d": [201, 143, 610, 997], "point_2d": [455, 445]} </answer> ✗

**Seg-ReSearch-8B (ours):**

<think> I need to identify the lead singer of the band that will release "Boycott Heaven" in 2026. I will search for information about the band and its lead singer. </think>
<search> {"name": "text_search", "query": "lead singer of the band releasing "Boycott Heaven" in 2026"} </search>
<information> Web Page 1: Nearly 20 years after The Format broke up, they're back with "Boycott Heaven." It's a reunion built on the friendship between **Nate Ruess** and ..., Web Page 2: In 2025, the band reunited for a second time and announced their third studio album, Boycott Heaven, for release in 2026... </information>
<think> I have found that the band is called The Format and that the lead singer is Nate Ruess. I will now search for an image of Nate Ruess to identify him in the video. </think>
<search> {"name": "image_search", "query": "Nate Ruess"} </search>
<information> 1. (Title: Nate Ruess | Spotify)

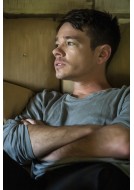

</information>
<think> I have obtained an image of Nate Ruess, the lead singer of The Format. I will now compare this image with the provided video frames to find the frame where he is most clearly visible. </think>
<keyframe> 2 </keyframe>
Here is the high-res version of your selected frame:

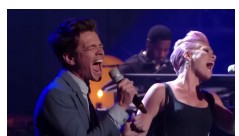

<think> In this frame, I can see a man with short brown hair, wearing a suit, singing into a microphone. His facial features and hairstyle match the image of Nate Ruess I found. </think>

<answer> {"bbox_2d": [120, 78, 607, 998], "point_2d": [275, 675]} </answer> ✔

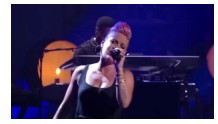 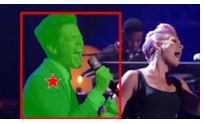 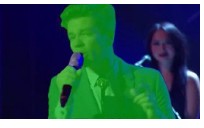 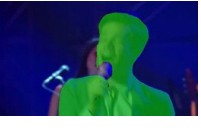 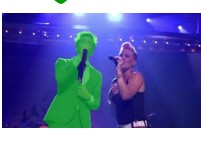 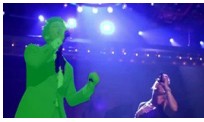

