# OpenReview forum: "Seg-ReSearch: Segmentation with Interleaved Reasoning and External Search"
_ICML.cc/2026/Conference — ICML 2026 regular_

### Official Review · Reviewer_Qfyh · 2026-02-28

**Soundness:** 2
**Presentation:** 3
**Significance:** 3
**Originality:** 2
**Overall Recommendation:** 4
**Confidence:** 3

**Summary:**

This paper presents Seg-ReSearch, a language-guided video object segmentation framework that integrates interleaved reasoning and external web search into a multimodal large language model (MLLM). The method enables multi-turn interactions with a search engine to retrieve external knowledge and localize target objects via a frozen segmentation model. The authors propose a hierarchical reward design that combines initial guidance, process-level incentives, and outcome-based localization rewards.

The paper also introduces OK-VOS, a new benchmark requiring outside knowledge for video object segmentation. Experimental results on OK-VOS and existing reasoning segmentation benchmarks demonstrate substantial improvements over search-augmented baselines.

**Compliance With Llm Reviewing Policy:**

Affirmed.

**Final Justification:**

The rebuttal has positively influenced my evaluation. In particular, the additional experiments on another backbone support generalization, and the controlled comparisons clarify that the performance gains mainly come from the learned reasoning policy rather than stronger retrieval. Additional evaluations also suggest robustness beyond the proposed dataset.

I still consider the methodological novelty to be somewhat limited, as the approach mainly integrates existing components with a task-specific reward design. However, given the importance of the problem, the strength of the empirical results, and the improvements provided in the rebuttal, I revise my recommendation to a weak accept.

**Key Questions For Authors:**

1. The evaluation is limited to Qwen3-VL models. Have you evaluated Seg-ReSearch on other open-source multimodal LLMs ?


2. Since OK-VOS is newly constructed in this work, have you considered evaluating on independently curated benchmarks that require external knowledge?

**Limitations:**

The discussion of technical limitations remains limited. For example, the dependence on external search engines, retrieval quality, and dataset-specific design choices are not thoroughly analyzed. A more detailed examination of failure cases, robustness issues, and broader deployment risks would strengthen the discussion of limitations.

**Strengths And Weaknesses:**

--Strengths:

1. The paper addresses a problem of incorporating external knowledge into language-guided video object segmentation.

2. The proposed framework is technically sound and well implemented, with clear reward definitions and comprehensive experiments.

3. The introduction of the OK-VOS benchmark highlights a challenging open-world setting.



--Weaknesses:

1. The methodological novelty is a bit limited, as the framework primarily integrates existing components (MLLM, search engine, RL fine-tuning, frozen segmentation model), with the main technical contribution lying in heuristic reward formulation rather than in new model architectures or learning algorithms. Also, the hierarchical reward design appears heuristic and task-specific.

2. Evaluation is restricted to a single backbone family (Qwen3-VL), making it unclear whether the improvements generalize to other multimodal LLMs.

3. Since OK-VOS is newly constructed by the authors, additional validation on independent benchmarks requiring external knowledge would strengthen confidence that the improvements are not tied to dataset-specific design choices.

4. The performance improvements are closely tied to retrieval configurations and search backends (e.g., varying Top-k entries, different search engines). While these experiments demonstrate the potential of enhanced retrieval, it remains unclear to what extent the reported gains come from the learned reasoning policy itself versus increased access to external information or stronger retrieval pipelines.

---

> ### Author Rebuttal · Authors · 2026-03-30
>
> We sincerely thank you for your time and suggestions to improve our manuscript. Below, we address your specific concerns.
>
> > **Q1. Clarifying the novelty.**
>
> **First**, we introduce a new task—segmentation with external knowledge, a challenge that existing methods completely fail to address. **Second**, we propose a new agentic segmentation framework with multi-round reasoning and search, which is a paradigm shift from existing segmentation pipelines that rely solely on single-round internal reasoning. **Third**, we propose a novel reward design that effectively balances external search and visual reasoning. This design can also provide insights for general agent training, which typically struggles with sparse outcome rewards or step-wise supervision. **Combining these efforts**, we believe the novelty of our work and its contributions to the field have been clearly demonstrated.
>
> > **Q2. Ablation of other MLLMs.**
>
> To address your concern, we apply our method to another backbone—Qwen2.5-VL-7B. As shown in Table R1, our method improves the baseline by +7.0 overall J&F, confirming that our method can generalize effectively to different MLLMs.
>
> | Method | One-hop | Multi-hop | Relational | Overall |
> | :--- | :---: | :---: | :---: | :---: |
> | `Qwen2.5-VL-7B*+Search` | 32.1 | 30.8 | 36.2 | 32.6 |
> | `+ Seg-ReSearch (Ours)` | 42.9 | 38.7 | 38.7 | 39.6 |
>
> Table R1. Overall performance with Qwen2.5-VL-7B backbones.
>
> > **Q3. Are the performance improvements tied to OK-VOS?**
>
> Since this is the first work to explore external knowledge in the detection/segmentation field, there are no existing benchmarks available to evaluate the search-augmented segmentation capability—that’s why we constructed OK-VOS. However, as shown in Tables 2 and 3, even on existing segmentation benchmarks that do not explicitly require external information, our method still improves the baselines consistently (+5.3% cIoU on ReasonSeg and +6.3% J&F on ReasonVOS). To further address your concerns, we conduct a cross-domain validation on Bamboogle, a challenging QA benchmark requiring external knowledge. As shown in Table R2, even without finetuning on QA datasets, Seg-ReSearch still achieves consistent performance improvements over the baseline. These results confirm the strong generalization and robustness of our method.
>
> | Method | Cover-EM | F1 |
> | :--- | :---: | :---: |
> | `Qwen3-VL-8B*+Search` | 47.2 | 53.6 |
> | `Seg-ReSearch-8B (ours)` | 63.2 | 66.2 |
>
> Table R2. Performance comparisons on Bamboogle benchmark.
>
> > **Q4. To what extent do the gains come from the learned policy versus external information/stronger retrieval pipelines.**
>
> In our experiment (Table 1), the baseline Qwen3-VL*+Search and our Seg-ReSearch shares the exact same prompt, search engine, and retrieval pipeline (including the number of retrieved passages and images). Therefore, the performance gap between Qwen3-VL*+Search and Seg-ReSearch naturally represents the gains (+12.4 J&F) from the learned policy.
>
> > **Q5. Add discussion of technical limitations.**
>
> We thank the reviewer for the constructive suggestion. We agree and will expand the Limitations section. Specifically, the performance of Seg-ReSearch is inherently influenced by the quality of search engines (e.g., their database coverage and ranking algorithms). For extremely long-tail concepts, the required information sometimes fails to appear in the top-K retrieved results, leading to segmentation failures. In the revision, we will provide detailed visualizations and discussion of such failure cases.

---

> > ### Author Rebuttal · Reviewer_Qfyh · 2026-04-03
> >
> > Thank you for the detailed rebuttal and additional experiments. The responses address several of my main concerns, and I adjust my score.

---

### Official Review · Reviewer_ByS3 · 2026-03-04

**Soundness:** 3
**Presentation:** 3
**Significance:** 3
**Originality:** 3
**Overall Recommendation:** 4
**Confidence:** 4

**Summary:**

The paper addresses the problem that user queries often involve up-to-date information or domain-specific concepts, which go beyond the “frozen” knowledge of Multimodal Large Language Models (MLLMs). It proposes Seg-ReSearch, which can perform interleaved reasoning and external search, and introduces a hierarchical reward mechanism to effectively resolve the tension between sparse outcome signals and overly rigid process supervision. In addition, the paper establishes OK-VOS, a human-annotated benchmark that explicitly requires external knowledge for video object segmentation.

**Compliance With Llm Reviewing Policy:**

Affirmed.

**Final Justification:**

The author has answered all my questions.

**Key Questions For Authors:**

(1) In the example shown in Fig. 1, the “Think” part in (b) for Seg-ReSearch identifies all frames that contain the target, whereas in (a) the “Think” part identifies only one frame and misses another frame that also contains the target. This example therefore makes Seg-ReSearch appear less robust.

(2) Please include a text-only retrieval ablation by setting the number of retrieved images to 0 (i.e., the “image=0” setting in Table 8), while keeping everything else unchanged. This is necessary to quantify how much of the gain comes from retrieved images (visual priors/template-like cues) . It also helps verify that Seg-ReSearch remains effective when no visually similar external image is retrieved, mitigating concerns about inflated performance due to external visual templates or leakage.

(3) Table 8 reports the default setting as Text = 3 and Image = 1, but in Fig. 1(a) the number of externally retrieved input images shown is four. Is this a mistake or an inconsistency?

**Limitations:**

yes

**Strengths And Weaknesses:**

Paper Strengths:

(1) For the open-world setting, the paper proposes Seg-ReSearch, a novel trainable agentic segmentation framework.

(2) The paper is generally well written and easy to follow. The motivation for streaming processing is clearly articulated, and the overall framework and individual components are well presented.

(3) Extensive experiments to validate the proposed approach.

Major Weaknesses:

(1) The core gains of this paper heavily rely on online Web search, which restricts the applicability of the method to settings where Internet access is available and external queries are permitted.

(2) Seg-ReSearch does not address genuinely unknown knowledge; rather, it shifts knowledge missing from a frozen MLLM to externally retrievable evidence. Consequently, its effectiveness critically depends on whether the target concept/fact can be retrieved from the web; if no such evidence exists, the method may degenerate into guessing.

(3) Since the method explicitly retrieves web images and feeds them into the policy MLLM, there is a non-trivial risk that retrieval returns near-duplicate depictions of the target object (or even frames from the same source video), effectively turning the task into template matching and inflating results.

---

> ### Author Rebuttal · Authors · 2026-03-30
>
> We sincerely thank the reviewer for the time and constructive feedback. We are encouraged by your recognition of our ***framework's novelty***, ***clear presentation***, and ***extensive experiments***. Below, we address your specific concerns.
>
>
> > **Q1. Reliance on Internet access.**
>
> We’d like to clarify that our core technical contribution lies in the interleaved reasoning-search policy, rather than the access of Internet. For most daily scenarios where Internet access is available, our method effectively expands the knowledge boundaries of MLLMs. For restricted offline environments, our method can also be seamlessly adapted by local retrieval system or private knowledge base (like our training setup).
>
> > **Q2. Addressing "genuinely unknown" knowledge.**
>
> Kindly note that we never claim Seg-ReSearch can resolve all unknown facts in the world. Previous segmentation methods rely entirely on the MLLMs’ frozen knowledge, which is bounded and could be outdated. In this work, we explicitly expand the MLLMs’ knowledge boundaries to external knowledge bases or even the entire Internet, where the knowledge is vast and daily updating. This paradigm shift from internal memory to external knowledge makes the detection/segmentation for long-tailed or new concepts possible. Therefore, we believe Seg-ReSearch has taken a meaningful step beyond existing works.
>
> > **Q3. Near-duplicate retrieved image.**
>
> To address your concern, we compute the similarity between the retrieved images and the selected keyframes. The average results (MSE: 0.07, SSIM: 0.14, CLIP: 0.46) prove they are not “near-duplicate”. Furthermore, we’d like to argue that the ability to actively search for images when encountering unknown concepts reflects the model's intelligence, rather than its weakness. Considering that the target name may not be available initially, the model must perform iterative reasoning and search to confirm the target identity before retrieving images, as illustrated in Appendix D. Therefore, this is a complex reasoning task rather than trivial “template matching”.
>
> > **Q4. Inconsistency of the “Think” parts in Figure 1 (a) and (b).**
>
> Kindly note that our system prompt only instructs the model to locate the target in a single clear frame, without requiring it to list all frames containing the target. In both Figure 1(a) and (b), Seg-ReSearch successfully identifies a valid keyframe and completes the task. It does not “miss” any step. During thinking, the MLLM can freely choose to mention multiple frames or focus on just one. This represents the inherent flexibility of MLLM reasoning, rather than a lack of robustness.
>
>
> > **Q5. Text-only retrieval ablation.**
> >
> As shown in Table R1, when removing image retrieval, the performance of Seg-ReSearch drops by 3.6 J&F, but still significantly outperforms the baseline Qwen3-VL-8B*+Search (using image retrieval) by 8.8 J&F. These results demonstrate that our improvements stem from a superior search policy rather than visual leakage. We will include these results in the revision.
>
> | Method | #image | J&F | J | F |
> | :--- | :---: | :---: | :---: | :---: |
> | `Qwen3-VL-8B*+Search` | 1 | 37.6 | 37.0 | 38.2 |
> | `Seg-ReSearch-8B (ours)` | 0 | 46.4 | 45.7 | 47.1 |
> | `Seg-ReSearch-8B (ours)` | 1 | 50.0 | 49.4 | 50.7 |
> |
>
> Table R1. Performance comparison with and without image retrieval.
>
> > **Q6. The number of retrieved images.**
>
> We'd like to clarify that Figure 1(a) is merely a qualitative example to illustrate our model's capability to autonomously search information or images. Actually, our framework is flexible and naturally supports multi-image retrieval. While the number of retrieved images is set to 1 by default, this hyper-parameter can be easily adjusted as needed.

---

> > ### Author Rebuttal · Reviewer_ByS3 · 2026-04-03
> >
> > The author has answered all my questions.

---

### Official Review · Reviewer_m8hL · 2026-03-12

**Soundness:** 3
**Presentation:** 4
**Significance:** 3
**Originality:** 3
**Overall Recommendation:** 4
**Confidence:** 4

**Summary:**

This paper proposes Seg-ReSearch, a novel segmentation framework that enables MLLMs to expand their knowledge boundaries through interleaved reasoning and external search, thereby performing segmentation for long-tail concepts and up-to-date information. The paper also introduces a hierarchical reward mechanism (combining Initial Guidance Reward, Tapering Process Reward, and Outcome Reward) to train this capability via reinforcement learning. Additionally, the authors establish OK-VOS, a new benchmark requiring outside knowledge for video object segmentation. Experiments show significant improvements over existing methods on OK-VOS and conventional reasoning segmentation benchmarks.

**Compliance With Llm Reviewing Policy:**

Affirmed.

**Final Justification:**

The authors have addressed all my concerns.

**Key Questions For Authors:**

1.	Could the authors report the inference time and computational overhead analysis mentioned above?
2.	Could the authors provide a sensitivity analysis for p in TPR?
3.	It would also help to discuss how often knowledge-intensive queries appear in practice and when Seg-ReSearch is needed versus standard approaches.

**Limitations:**

The authors discussed potential negative societal impact of their work but did not discuss the limitations.

**Strengths And Weaknesses:**

Strengths:

1.	The paper addresses an important limitation of existing segmentation systems—their reliance on frozen internal knowledge. The focus on handling up-to-date information and long-tail concepts is timely and relevant for real-world applications.

2.	OK-VOS is a carefully constructed benchmark with rigorous human annotation and multi-round review to ensure queries genuinely require external knowledge. The three-tier categorization (One-hop, Multi-hop, Relational) provides nuanced evaluation.

3.	Seg-ReSearch achieves substantial improvements (+10% over search-augmented baselines on OK-VOS) and establishes new SOTA on conventional benchmarks (ReasonSeg, ReasonVOS).

Weaknesses:

1.	This paper lacks analysis of the inference latency and computational overhead introduced by external search. Given that the model requires multi-turn search iterations and external API calls, the accuracy-efficiency trade-off is critical for practical deployment. The paper should report average end-to-end inference time and compare it with baselines.

2.	The hyperparameter p in the Tapering Process Reward (TPR) critically controls the trade-off between exploration steps and performance gains. However, the paper only illustrates reward curves for p=0.7 and p=0.3 in Figure3. Could the authors provide a more comprehensive sensitivity analysis table comparing overall J&F performance under different p values (e.g., p = {0.5, 0.7, 0.9})? This would help verify the robustness of the chosen hyperparameter setting.

3.	Although the paper presents Seg-ReSearch as a solution for segmentation tasks involving long-tail concepts or very recent information that may require external knowledge, I’m not entirely convinced how common these cases are in real-world use. It would be helpful if the authors could provide some empirical evidence—for example, user studies, analyses of existing dataset logs, or statistics from real deployment scenarios—to show that these kinds of knowledge-intensive segmentation queries actually occur with some frequency. Otherwise, it’s a bit hard to judge whether the added complexity of interleaving search is really necessary. Without such evidence, the problem setup, while interesting from a technical perspective, may feel somewhat constructed rather than clearly driven by practical user needs.

4.	While the specific formulation of the Tapering Process Reward (TPR) is novel, the core concept of hierarchical rewards is well-established in reinforcement learning literature. The paper should more precisely position its main contribution as the novel TPR formulation and its application to search-augmented segmentation, rather than presenting it as a fundamentally new reward mechanism. This would provide a more accurate and rigorous characterization of the methodological innovation.

---

> ### Author Rebuttal · Authors · 2026-03-30
>
> Thanks for your thoughtful and encouraging comments. We provide our repsonses as follows.
>
> > **Q1. Comparison of inference time.**
>
> As shown in Table R1, for queries that do not require external knowledge (e.g., ReasonSeg and ReasonVOS), Seg-ReSearch only brings a negligible extra time burden (<1s), as very few external searches are called. For complex queries (e.g., in OK-VOS) that require external knowledge, the multi-round search does incur extra latency (~2s). However, this extra time enables our method to successfully solve challenging tasks that existing methods completely fail to address (with ~15% improvements in overall J&F). Therefore, we believe Seg-ReSearch has achieved a reasonable trade-off between efficiency and performance gains.
>
> | Method | ReasonSeg | ReasonVOS | OK-VOS |
> | :--- | :---: | :---: | :---: |
> | `Qwen3-VL-8B*`|3.2|1.7|6.1|
> | `Seg-ReSearch-8B`|4.1|1.8|8.0|
>
> Table R1. Average time cost (*sec.* / *sample*) on various benchmarks, where Qwen3-VL-8B\* is the single-round baseline without external search.
>
> > **Q2. Ablation of $p$ values.**
>
> As suggested, we provide the results across various $p$ in Table R2. A larger $p$ implies a smaller bonus for extra search. With $p$ increasing from 0.2 to 0.7, the overall J&F increases from 44.4 to 46.0. This reflects a trade-off between extensive exploration and precise search. However, when $p$ is further increased to 0.9, the bonus becomes too marginal to encourage necessary search, causing a sharp performance drop to 41.0. We will include these results in the revision.
>
> | p | J&F | J | F |
> | :---: | :---: | :---: | :---: |
> | 0.2 | 44.4 | 43.7 | 45.1 |
> | 0.5 | 45.3 | 44.8 | 45.9 |
> | 0.7 | 46.0 | 45.3 | 46.7 |
> | 0.9 | 41.0 | 40.3 | 41.6 |
>
> Table R2. Overall performance across different $p$ value.
>
> > **Q3. Statistics of real-world demand for knowledge-intensive visual queries.**
>
> To address your concern, we turn to *WildVision-Chat*, which contains over 46K real-world multimodal conversations. *These data are sourced from an online platform where users upload their own images and freely query top-tier MLLMs* (e.g., GPT-4o, Claude, Gemini). We utilize Qwen3.5 to evaluate the frequency of these queries requiring external knowledge (e.g., asking about long-tail entities, up-to-date information, or domain-specific concepts). The results reveal that 23.5% of the user queries are knowledge-intensive. This proportion reflects the critical demand of real-world user queries for external knowledge.
>
> > **Q4. Position the novel TPR formulation as the main contribution.**
>
> Thanks for your constructive suggestion. We will reclaim our contribution regarding the reward design as follows: *We formulate a novel Tapering Process Reward (TPR) for search-augmented segmentation, which effectively balances the policy learning between external search and visual reasoning.*

---

> > ### Author Rebuttal · Reviewer_m8hL · 2026-04-05
> >
> > The authors’ rebuttal on the necessity of the proposed framework is not sufficiently convincing. In particular, when viewed through XciW’s Q1, the experiments look more like author-controlled constructive validation than evidence that the “decomposition + external retrieval–driven pipeline” reflects a realistic user need distribution. More importantly, the current evidence does not convincingly show that this pipeline is truly necessary in real-world settings or that users would benefit from it. As a result, even with improved results, it’s hard to argue that this work addresses a real and critical gap.

---

> > > ### Author Response · Authors · 2026-04-05
> > >
> > > We sincerely thank you for acknowledging our responses regarding time cost and ablations. Here, we'd like to further address your concerns regarding the real-world necessity.
> > >
> > > **1. Addressing a real and critical gap**
> > >
> > > As stated in the paper, **the primary goal of this work is to leverage external knowledge to segment new or long-tail concepts** (e.g., *segmenting the new iPhone 17*). This is a common and realistic demand, yet none of the existing methods can handle it. New concepts, products, and events emerge daily. Without external search capabilities, current segmentation models cannot address real user needs. In contrast, our proposed method not only effectively addresses this new challenge on our benchmark (Table 1) but also consistently improves SOTA performance on existing segmentation benchmarks (Tables 2 & 3). These consistent improvements across various scenarios clearly demonstrate the practical value of our approach.
> > >
> > > **2. The value of sequential search**
> > >
> > > Multi-hop search is indeed a real demand in real-world scenarios. In practice, user queries are often ambiguous and fragmented, making single-hop search insufficient to handle various cases. In our paper, we provide extensive quantitative comparisons and qualitative visualizations to demonstrate that our Seg-ReSearch has effectively learned not only to automatically search for unknown concepts, but also to perform sequential search when encountering multi-hop queries. We believe these capabilities are essential for a truly open-world detection/segmentation system. We also acknowledge that, in order to eliminate visual and linguistic shortcuts, the current version of benchmark does not perfectly align with the real user query distributions. As promised in our previous response, we will continue to include more common queries in future versions to further improve the benchmark.
> > >
> > > We hope these explanations can address your remaining concerns.

---

### Official Review · Reviewer_XciW · 2026-03-16

**Soundness:** 3
**Presentation:** 3
**Significance:** 3
**Originality:** 2
**Overall Recommendation:** 4
**Confidence:** 3

**Summary:**

This paper proposes Seg-ReSearch, a framework for video/image segmentation that integrates interleaved reasoning and external search. The motivation is that many real-world queries depend on external or up-to-date knowledge that is not contained in the frozen parameters of MLLMs. To address this limitation, the system allows the model to iteratively perform reasoning and issue search queries to retrieve textual or visual information before localizing the target object. To train this capability, the authors introduce a hierarchical reinforcement learning reward design that balances exploration and task performance. The paper also introduces OK-VOS, a new benchmark for video object segmentation that explicitly requires outside knowledge.

**Compliance With Llm Reviewing Policy:**

Affirmed.

**Final Justification:**

My concern is mostly addressed so I retain my previous positive rating.

**Key Questions For Authors:**

Please see above weakness.
Also it would be helpful if the authors can provide training data examples.

**Limitations:**

Yes.

**Strengths And Weaknesses:**

Strengths

1. The paper is well written and easy to follow. The motivation for each system component is clearly explained.

2. The paper proposes a simple RL framework to improve the model’s ability to perform multi-step reasoning and search interactions before segmentation.

3. The proposed OK-VOS benchmark evaluates segmentation systems in scenarios requiring external knowledge.

Weaknesses / Questions

1. Some relational questions in OK-VOS appear somewhat artificial and may not resemble how users typically refer to objects in visual content. For example, queries such as “the artist who took the top prize at the American Music Awards several months prior to the public release of Nano Banana” combine multiple pieces of unrelated information. It would be helpful if the authors could clarify the intended use cases for such queries and whether they reflect realistic user scenarios.

2. The method performs multi-step reasoning and search interactions before segmentation. It would be helpful to include a time cost or latency analysis (e.g., on ReasonSeg and ReasonVOS) to better understand the time cost.

3. The paper reports results for Qwen3-VL + Search. Can the authors provide example on how the baseline is prompted or informed that search is available? Clarifying the prompt format and how search actions are triggered would help ensure a fair comparison.

4. The benchmark is designed so that queries require knowledge beyond the internal knowledge of current models. However, as future models are trained on more recent data, some queries may become answerable without external search. It would be helpful if the authors could discuss how the benchmark remains meaningful as models evolve.

---

> ### Author Rebuttal · Authors · 2026-03-30
>
> Thank you for your constructive feedback and appreciation of our work. We provide our repsonses as follows.
>
>
> > **Q1. Clarification on intended use cases.**
>
> OK-VOS contains three types of queries, aiming to reflect realistic user demands from different aspects:
> - **One-hop:** A common need to segment unknown or new concepts (e.g., *Labubu*, *iPhone 17E*).
> - **Multi-hop:** In reality, users often forget specific object names and can only provide fragmented clues (e.g., *referring to an actor but only remembering some movies he has starred in or some awards he has won*). This requires the ability to decompose complex expressions and conduct sequential searching (as in your example).
> - **Relational:** This reflects the need to combine searched knowledge with the visual content (e.g., *segment the toy appearing before Labubu in the video*).
>
> As the first benchmark for this task, OK-VOS must first ensure a rigorous and comprehensive evaluation of the autonomous search ability. To this end, we explicitly integrated diverse, up-to-date information to construct the search chains, ensuring models cannot simply "guess" answers without conducting sequential searches. However, we acknowledge that this rigorous design could make some queries less common. We appreciate your suggestions and will include more common queries in future benchmark updates.
>
> > **Q2. Time cost.**
>
> As shown in Table R1, for queries that do not require external knowledge (e.g., ReasonSeg and ReasonVOS), Seg-ReSearch only brings a negligible extra time burden (<1s), as very few external searches are called. For complex queries (e.g., in OK-VOS) that require external knowledge, the multi-round search does incur extra latency (~2s). However, this extra time enables our method to successfully solve challenging tasks that existing methods completely fail to address (with ~15% improvements in overall J&F). Therefore, we believe Seg-ReSearch has achieved a reasonable trade-off between efficiency and performance gains.
>
> | Method | ReasonSeg | ReasonVOS | OK-VOS |
> | :--- | :---: | :---: | :---: |
> | `Qwen3-VL-8B*`|3.2|1.7|6.1|
> | `Seg-ReSearch-8B`|4.1|1.8|8.0|
>
> Table R1. Average time cost (*sec.* / *sample*) on various benchmarks, where Qwen3-VL-8B\* is the single-round baseline without external search.
>
> > **Q3. Prompt for the Baseline.**
>
> For fair comparison, the baseline *Qwen3-VL\* + Search* and our Seg-ReSearch share the **same prompt**, which is detailed in Appendix A (page 12). Specifically, we provide the definition and calling format of the search tool, and instruct the models to call it whenever external knowledge is required. Furthermore, other configurations (e.g., search engines, retrieval pipelines, and maximum search turns) are also kept consistent for fair comparison.
>
> > **Q4. How the benchmark remains meaningful as models evolve.**
>
> We appreciate this forward-looking perspective. We’d like to discuss it in two aspects. **First, in addition to new information, OK-VOS also features vast long-tail knowledge** (e.g., *who served as the Member of Parliament for Mole Valley from 1983 to 1997*). Even as future models are trained on more recent data, it remains highly challenging to memorize all long-tail facts. **Second, we believe that benchmarks and models are co-evolving.** We view OK-VOS as a long-term evolving plan and the current version represents the first step toward external-knowledge-driven segmentation. In the future, we will not only iteratively update the benchmark but also explore automatic data construction pipelines. We hope this response can address your concern.
>
> > **Q5. Training data examples.**
> >
> Our training set covers four types of queries: one-hop (15%), multi-hop (40%), relational (25%), and no-search (20%). Representative examples are as follows:
> - **One hop:** *The actor who voiced Danny Wegbreit in Long Story Short.*
> - **Multi-hop:** *The actor whose father was convicted of assassinating federal judge John H. Wood Jr.*
> - **Relational:** *The player who receives the ball from the winner of 2018 European Golden Ball.* (from a football match clip)
> - **No-search:** *Which person is most likely to win first place?* (from a running race video)
>
> We will include these in the revision.

---

> > ### Author Rebuttal · Reviewer_XciW · 2026-04-05
> >
> > I think the authors address most of my concern. But I do think it is important that the authors include more common queries in future benchmark updates and also keep updating the benchmark when necessary.

---

### Decision · Program_Chairs · 2026-04-30

**Decision:**

Accept (regular)

**Comment:**

This paper proposes Seg-ReSearch, a segmentation approach that interleaves reasoning with external search to progressively refine segmentation via retrieval-augmented reasoning.

Some of the noted strengths are: an interesting, search-integrated segmentation pipeline; multi-round agentic reasoning with external knowledge; strong generalization across multiple backbones and domains; and a principled reward design that balances search & reasoning. The robustness and generalization evidence is also convincing. Weak points highlighted include: methodology mostly integrates existing components; performance is inherently tied to retrieval quality (long-tail concepts may fail).

All reviews converged to accept after a productive discussion. AC agrees with the consensus and recommends an Accept decision.